# “We Live Our Life Normal”: A Qualitative Analysis of Nigerian Women’s Health-Seeking Behavior during the COVID-19 Pandemic

**DOI:** 10.3390/ijerph21030265

**Published:** 2024-02-24

**Authors:** Mary Ndu, Gail Teachman, Janet Martin, Elysee Nouvet

**Affiliations:** 1Department of Health and Rehabilitation Sciences, Faculty of Health Science, Western University, 1151 Richmond St, London, ON N6A 3K7, Canada; 2Faculty of Health Science, School of Occupational Therapy, Western University, 1151 Richmond St, London, ON N6A 3K7, Canada; gteachma@uwo.ca; 3Departments of Anesthesia & Perioperative Medicine and Epidemiology & Biostatistics, Schulich School of Medicine & Dentistry, Western University, 1151 Richmond St, London, ON N6A 3K7, Canada; 4School of Health Studies, Faculty of Health Science, Western University, 1151 Richmond St, London, ON N6A 3K7, Canada; enouvet@uwo.ca

**Keywords:** COVID-19, interpretive phenomenological analysis, Nigeria, global health, social determinant of health, lived experiences

## Abstract

Background: This study examined where women sought healthcare during the COVID-19 pandemic and their reasons for doing so. We aim to understand further how women accessed care during the COVID-19 pandemic to inform future preparedness and response efforts. This knowledge gained from this study can inform strategies to address existing gaps in access and ensure that women’s health needs are adequately considered during emergencies. Methods: This study used an interpretive phenomenological-analysis approach to analyze data on women’s experiences with healthcare in Nigeria as the COVID-19 pandemic progressed. Semi-structured interviews were conducted with 24 women aged 15 to 49 between August and November 2022 and were supplemented with three focus-group discussions. Results: Following our analysis, three superordinate themes emerged: (i) barriers to seeking timely and appropriate healthcare care, (ii) the influence of diverse health practices and beliefs on health-seeking behavior, and (iii) gendered notions of responsibility and of coping with financial challenges. Conclusions: This paper examined women’s decision to seek or not seek care, the type of care they received, and where they went for care. Women felt that the COVID-19 pandemic affected their decision to seek or not seek care.

## 1. Introduction

Timely and appropriate services are critical to maintaining and sustaining the health of women and children. In the last decade, a consistent focus has been on improving health services as a fundamental approach to minimizing mortality from preventable causes in Nigeria and the sub-Saharan region [1,2]. Emphasis has been on pregnant and breastfeeding mothers, who constitute the highest proportion of global deaths from preventable causes, given the complicated cascade of events resulting in pre- and postpartum deaths [3]. Nigeria, for instance, contributed 14% of the global maternal deaths in 2020, with over 58,000 women dying pregnancy-related deaths, one of the highest maternal mortality rates globally [4,5]. The Nigeria Centre for Disease Control and Prevention (NCDC) December 2021 data showed that women had less mortality from COVID-19 compared to men [6]. However, women were still disproportionately affected by the COVID-19 pandemic [7,8].

At the onset of the COVID-19 pandemic, evidence showed that pandemic preparedness and response strategies must be tailored to local contexts, considering factors such as population density, access to healthcare, and social determinants of health [9]. They also emphasize the importance of community engagement and participation in pandemic response efforts and the need for global cooperation and solidarity to ensure equitable access to vaccines and other essential health resources. Therefore, it is crucial to redefine the equity agenda of global health beyond analyzing the determinants of COVID-19 and to include scrutiny of the short- and long-term effects of public health measures implemented in response.

The COVID-19 pandemic had a far-reaching global impact with overwhelming effects on health systems, especially in countries like Nigeria with poorly managed health systems. Like many other sub-Saharan African countries, Nigeria faces considerable impediments to women’s access to healthcare owing to factors such as poverty, limited education, and cultural norms prioritizing men over women [4,10,11,12,13]. The challenge of accessing health care has plagued Nigeria since before the COVID-19 pandemic, with several studies echoing women’s health behavior fluctuating between biomedical and non-biomedical care [8,13,14]. These challenges became more apparent in light of the one-size-fits-all response approach adopted at the onset of the COVID-19 pandemic despite variations in the severity of its impact on individual countries [15]. The resulting barriers were highlighted and further compounded by the COVID-19 pandemic, which resulted in disruptions in healthcare systems, economic downturns, increased household stress and burdens, and amplified disparities within and between nations [9,16,17]. Current studies indicate that Nigerian women had difficulty accessing health services with a significant reduction in essential routine health services, including maternal and child health services [14,18,19]. These declines in healthcare utilization during the COVID-19 pandemic amplified the COVID-19 pandemic’s harmful effects on health outcomes [19,20]. Decreased health-service usage provides significant insight into women’s health-seeking behavior. Such a decline holds the potential to reverse the gains made in reducing maternal and child mortality in Nigeria in the last decade [19].

Existing frameworks on health-seeking behavior posit that disruptive factors within the environment can influence women’s health-seeking behavior, affecting their health-services use [21,22]. For instance, the three-delay model noted that women’s hesitation to seek care results from barriers such as inadequate transportation, leading to women arriving late to the hospital, long travel distances to get to the hospital, and poor awareness of health issues [23]. Similarly, Andersen’s health-utilization framework posits that societal factors and the broader environment combine to influence a person’s decision to seek care [24]. The transtheoretical model conceptualized health behavior as an intentional process of action in which a person can choose to engage [25]. Current studies on women’s health-seeking behavior during COVID-19 show that negative experiences with the health system during disruptions affected their intentions to seek care [23,26,27]. However, while we recognize that systemic issues drive women’s health-seeking behavior, it is also important to note that personal health beliefs and perceptions of health care are significant contributors to women’s intentions to use health services [19,28]. In this paper, we acknowledge that there are several pathways to health, and each woman is unique in their approach to it. In recognition of the plurality of experiences and approaches to health, no one framework or model is sacrosanct to describing the different patterns of women’s health behavior. As such, this study is grounded in the concept of health system responsiveness, which refers to the ability of the health system to respond to the population’s expectations based on their experiences of interacting with the health system [29].

This paper is a component of a larger study which aimed to analyze women’s experiences with healthcare in three states of Nigeria during the COVID-19 pandemic, as these experiences have implications for future pandemic preparedness, planning, and national health policy. It answers the research question, “Where and why did women in Nigeria seek or delay health care for themselves or their children during the COVID-19 pandemic?” By examining where women sought healthcare during the COVID-19 pandemic and their reasons for doing so, this paper sheds light on the implications of the COVID-19 pandemic for women’s health in Nigeria and globally. This paper can help identify and address women’s healthcare barriers in sub-Saharan African countries like Nigeria, such as poverty, a lack of education, and cultural attitudes. Recognizing how such factors intersect with women’s health-seeking behavior is important for promoting health equity, a fundamental principle of global health [30,31]. Understanding how women accessed healthcare during the COVID-19 pandemic can inform future pandemic preparedness and response efforts. This knowledge can be used to address gaps in access and ensure that women’s health needs are addressed during emergencies.

## 2. Materials and Methods

### 2.1. Study Setting

This research, conducted in purposively selected rural and semi-urban communities located in three states, namely Sokoto (northwest), Ebonyi (southeast), and Ogun (southwest), ensured the representation of diverse experiences of women from distinctive regions in Nigeria. In each state, we sampled women in one local government selected based on their location for security, given the widespread insecurity in Nigeria. Each local government combines a mix of communities identified as rural and semi-urban. The selection of these three states provided an opportunity for a comprehensive analysis of various social and cultural factors that may shape health-seeking practices. Nigeria is a culturally diverse country with over 300 distinct languages and ethnic groups, each with recognized languages, traditions, nuances, and norms that make them distinctive. In the north, where Sokoto State is located, Hausa and Fulani are the dominant languages. In the west, where Ogun State is located, the predominant language is Yoruba. In the eastern region, the Igbo language is the dominant language. The study locations had primary health care centers, with local pharmacies and private hospitals, except for Sokoto, which only had a primary health care center. The services available to women through these sites ranged from routine services, i.e., malaria treatment and immunization, to delivery. The private hospitals also offered surgical services (i.e., Caesarean section).

### 2.2. Study Design

Interpretative phenomenological analysis (IPA) was the research methodology used in this study. IPA’s interpretative and exploratory approach is particularly useful when examining complex and impactful lived experiences, such as women’s experiences with healthcare during the COVID-19 pandemic in Nigeria [32]. IPA is unique in its theoretical underpinning, objective, and processes. It aims to identify areas of convergence and divergence among participants with shared experiences [32].

In comparison to other qualitative research methods, such as narrative inquiry, which emphasizes the construction of meaning through storytelling and how people make meaning through stories, IPA is interested in understanding the meaning people ascribe to their experiences within the context of their personal and social worlds [32,33]. The process of analysis also differs significantly from other forms of qualitative analysis. While researchers using many different qualitative research approaches can code using qualitative software for analysis, IPA requires that researchers submerge themselves in the transcript by reading and re-reading transcripts, manually noting the discourse and emerging codes.

As an interpretative approach, IPA recognizes that researchers’ interpretations play a vital role in the analysis process, resulting in a double hermeneutic [34]. Thus, IPA aims to produce in-depth interpretative accounts of a small number of participants rather than a generalizable account for a larger sample, as it recognizes that each participant’s experience is unique and cannot be generalized to others [35]. By focusing on a small number of participants and producing rich and detailed accounts of their experiences, IPA can provide valuable insights into the lived experiences of these women and help identify areas where healthcare services can be improved to meet their needs better. Given IPA’s aim to explain subjects’ experiences separately from the whole, it is appropriate to show how global events can affect people differently. It recognizes the plurality of experiences, even in shared events like the global pandemic.

To answer the research question, “Where and why did women in Nigeria seek or delay health care for themselves or their children during the COVID-19 pandemic?” we asked women about their behavior during the COVID-19 pandemic when they or their child was sick, focusing on the period spanning the lockdown and the total and partial restrictions of interstate and intrastate travel. As such, we designed in-depth interviews and focus-group discussion guides with open-ended questions to generate information on women’s experiences with healthcare-seeking at different temporal points—before the COVID-19 pandemic, during the restriction of movement from March 2020 to July 2020, and after the restriction of movement. These questions invited women to recall details of their healthcare-seeking decisions in each of these periods, elaborate on factors they identified as influencing whether, when, and where they sought care, and if they sought out care, to share details of their encounters with healthcare services. Before the study commenced, the lead researcher piloted the tool in one study location to refine and ensure the prompts were clear and understandable. The in-depth interviews and focus-group discussions were in the dominant local language of the data collection site. Professional language translators from the Nigeria Translators Association translated the tools. 

### 2.3. Participants and Recruitment

In recruiting women for the focus-group discussions, we targeted women with varied health experiences during the COVID-19 pandemic. With the assistance of a local partner, we conducted outreach to local women’s groups in the community. The research assistants, resident in the study locations, were the local mobilizers of the women in the communities. We aimed to recruit women involved in making health decisions for themselves and their children. At the outreach meeting, we screened women based on their self-identification as mothers, socioeconomic status, and health needs during the COVID-19 pandemic. Although socioeconomic status was not a criterion, we paid attention to ensure we had a broad spectrum of women. However, we prioritized women who were pregnant during the COVID-19 pandemic or had a child. We focused on hearing from women who gave reasons for seeking and not seeking care during these periods. For the in-depth interview, we aimed to purposively select women who would provide rich and robust data for analysis. We were also attentive to recruiting women who made insightful comments that required further probing.

We also used snowballing, where women who attended the focus-group discussions informed their friends and neighbors about the study to increase our pool of women for in-depth interviews. These women were provided with the mobile numbers of the research assistants whom they contacted to indicate interest in participating in the interview. The research assistants then screened the women, prioritizing their status as mothers. Following this contact, the research assistants and the lead researcher fixed a date and time to meet with the women at a convenient place. We purposively selected twenty of the forty women invited to the focus-group discussion for in-depth interviews. The remaining four women were from referrals.

In using IPA, researchers are advised to use a small sample size, given its idiographic commitment [36]. While there is no correct answer to the question of sample size in IPA, 4–15 participants is the recommended size. Our aim to generate rich and compelling data representing multiple perspectives within a heterogeneous population guided our decision to recruit a larger sample. Given that data collection occurred across three sites, detailing some women’s personal experiences, this number is small to make inferences on the broader population of Nigerian women based on the findings because of the diversity of women in the study context.

### 2.4. Data Collection

Before the study commenced, the lead researcher piloted the tool in one study location to refine and ensure the prompts were clear and understandable. The lead researcher and the three research assistants conducted the focus-group discussions with alternating roles as note taker and recorder. After each focus group, the lead researcher and the research assistants debriefed on observations of the non-verbal information from each session, which the lead researcher recorded in her filed notes. Each focus-group discussion lasted for approximately an hour and thirty minutes. The study conducted twenty-four interviews and three focus-group discussions of 10–12 women each between August and November 2022. The first author was responsible for the interviews with the support of three Nigerian research assistants with prior qualitative research experience. The lead researcher in Nigeria is Igbo and speaks Yoruba with an understanding of Hausa. However, while the lead researcher led the interview in Ebonyi, to avoid any cultural complexities or challenges in the study, the research assistants from Ogun and Sokoto led the interviews of the women in those locations. Each interview lasted approximately 45 to 60 min. The team used eight open-ended questions as a guiding structure for the interviews. At the initial meeting introducing the study, the women invited to participate in the focus groups provided written consent which was reconfirmed verbally during the focus-group discussions and interviews. All names used in this study pseudonyms chosen by the women.

### 2.5. Data Analysis

Before commencing the fieldwork, the tools were transcribed to local languages using translators from the national translators’ association into Igbo, Hausa, and Yoruba. The lead translator confirmed the accuracy of the translations before the interviews. After the interviews and focus-group discussions, the research assistants, all indigenes of the data collection sites, trained to conduct interviews in the local languages, transcribed the transcripts. External readers reviewed and verified the final translated transcripts for accuracy. During translation, the emphasis was on highlighting unique traditional descriptions of ailments. We used a close term in cases where no direct translation of English terms was possible. For instance, we used headache, ori fifo, which loosely translates to one’s head being cut off.

The lead researcher initiated the data-coding process following IPA-coding guidelines [33]. The guidelines emphasize reflexivity to facilitate the researcher’s exploration of her presumptions, preconceptions, biases, and emotional responses. Through the repeated reading of each interview, the lead researcher identified codes and patterns of response, critically assessing her interpretations, and returning to the recorded interview where necessary to ensure an ethical representation of the emotions and expressions of the participants [32,33]. Subsequently, the lead researcher collated the themes and subthemes that emerged from the coding process, creating a comprehensive framework that captured the women’s experiences. Throughout the analysis, the principal investigator engaged in ongoing discussions with EN, JM, and GT, ensuring that the themes and subthemes were traceable to the participants and that emerging patterns were systematically identified and recorded. This methodological approach strengthened the overall credibility of the study [37].

## 3. Results

In this paper, a subset of transcripts from 14 women (see Table 1 and Table 2) was analyzed, described, and interpreted to highlight the emergent themes. The larger study sample was comprised of women aged 15 to 49. A total of 24 women—eight from each state—with diverse experiences and socioeconomic backgrounds participated in the study. Specifically, thirteen women had completed some secondary education, five had a college education, and six had obtained a university degree. Of the thirteen women who had completed secondary education, eight were unemployed. The remaining 18 women held various occupations, including teaching, catering, community health work, and civil service. The women were from different socioeconomic backgrounds, with some facing constraints to meet their basic daily needs. Most women were married, with only two exceptions: one a divorcee and the other a widow. Women lived far from the closest hospitals and primary health care centers. While several women reported travelling for up to an hour to access the nearest referral hospital, many reported living within walking distance of the nearest primary healthcare center. Women in the study reported using various healthcare resources, including professional contacts and traditional remedies. Of the 24 women interviewed in the large study, 20 had some form of interaction with the health system during the COVID-19 pandemic. At the same time, the remaining four indicated they had no reason to visit any hospital. The findings suggest that despite having different cultural distinctions, there was no significant difference in their experiences and responses to the COVID-19 pandemic.

In what follows, we present three interconnected superordinate themes that emerged from some women’s accounts of their health-seeking behaviors during the COVID-19 pandemic: (i) barriers to seeking timely and appropriate healthcare care, (ii) the influence of diverse health practices and beliefs on health-seeking behavior, (iii) and gendered notions of responsibility and of coping with financial challenges. These themes highlight an interplay between contextual conditions, norms, ideals, and women’s decisions of when, why, and from whom they sought healthcare during the COVID-19 pandemic. These findings reveal that where and why women in Nigeria sought or delayed health care for themselves or their children during the COVID-19 pandemic varied depending on the severity of the illness, social connections around their immediate environments, understanding of the COVID-19 pandemic, and personal beliefs about health in general. The overarching themes from our analysis are presented using an interpretive approach.

### 3.1. Barriers to Seeking Timely and Appropriate Health Care

This theme explored the systemic factors that influenced women’s decision making regarding healthcare-seeking behaviors during the COVID-19 pandemic. More specifically, it presents the various barriers women reported to have encountered within the health ecosystem, ultimately impacting their willingness to seek care. The women’s journey to access health care was often fraught with obstacles, significantly affecting their ability to receive timely and appropriate medical attention. These barriers were multifaceted, encompassing socioeconomic, cultural, systemic, and personal factors. The subthemes below present specific factors that contributed to this theme, including extracts to support the factors.

#### 3.1.1. Potential Fear of COVID-19 (mis)Diagnosis

This subtheme highlights how the fear of COVID-19 may prevent women from seeking necessary medical attention, even for non-COVID-related illnesses. It also includes the emotional and psychological impacts of the COVID-19 pandemic beyond just the physical health consequences, underscoring the importance of continuously educating the public about infectious disease symptoms and addressing misinformation through health literacy campaigns. Some pregnant women felt that if they sought care from a hospital, they might be subjected to stigma and misunderstanding due to their pregnancy symptoms, which some perceived could be mistaken for signs of COVID-19. They reported feeling weak and experiencing sneezing, which was sometimes misinterpreted by others as signs of the COVID-19 virus. For instance, Bimbo expressed concern about being subjected to stigma and discrimination:
“*People mistake that [pregnancy] weakness for a sign of COVID-19 if we sneeze, some people assume we have the virus*”.[Bimbo, Sokoto]

Her statement highlighted the emotional impacts of the COVID-19 pandemic. Her behavior reflected a potential to self-isolate, leading to loneliness, as she wanted to avoid going out and interacting with others. Potentially, the first thought that came to mind when feeling ill was whether she had the COVID-19 virus.
“*When you start feeling sick, the first thing that comes to mind is whether you have COVID-19. You don’t want to go out and have people say that you have it*”.[Bimbo, Sokoto]

Her perceptions could suggest an internalized stigma from past experiences dealing with health workers and society during epidemics or other infectious disease outbreaks, including that of the Ebola virus. The fear of being wrongly associated with the virus prevented women like Bimbo from interacting with others. No woman wanted to be identified as having had the virus. An experience that should have been celebrated appeared to have left some women feeling anxious about exposing themselves or their loved ones to the virus. This anxiety about the stigma associated with COVID-19 had women concerned about being labeled COVID-19 patients due to social implications, such as being judged. This often affected their decision to seek care when ill during the COVID-19 pandemic. For instance, some women worried about taking their child to the hospital because they were concerned that the child might be diagnosed with COVID-19. Instead, they chose to use a chemist (local medicine vendor).
“*During the COVID-19 pandemic, my child had a cold, but I was afraid to take him to the hospital. I was afraid to take him to the hospital, fearing that they would say it was “coro. I took him to a chemist. After two days, there was no improvement, so I took him to the hospital*”.[Ibironke, Ogun]

Similarly, Chioma’s statement showed a complex interaction between a mistrust of health workers, a fear of contracting the virus, and self-reliance in healthcare behavior during the COVID-19 pandemic.
“*During COVID-19, anytime I am sick, I used to be afraid of going to the hospital so that they would not say I have coronavirus. So, I take Panadol*”.[Chioma, Ebonyi]

Her fear could be interpreted in two ways: as a fear of contracting the virus and as a fear of being incorrectly labelled as a COVID-19 patient. Like many women in this study, her fear reflected a possible mistrust or unhealthy skepticism of the health system. Her behavior, like Ibironke and Bimbo, also resonated with public sentiments during the COVID-19 pandemic. Chioma’s behavior could stem from her health beliefs and the overwhelming presence of COVID-19 concerns that dominated the healthcare system during the study period. Her final statement, “So, I take Panadol.” can be interpreted as an attempt to maintain some internal locus of control over her personal health, given the heightened risks of contracting COVID-19. Bimbo, Chioma, and Ibironke’s behaviors are further illustrated by Halima’s statement:
“*One of the major challenges was visiting a doctor whenever I have a cold or Catarrh [excessive buildup of mucus in the nose or throat] because of fear of being diagnosed with COVID-19*”.[Halima, Sokoto]

These perceptions could suggest an internalized stigma from past experiences dealing with health workers and society during epidemics or other infectious disease outbreaks, including that of the Ebola virus disease. For women, the perception that any sign of illness, such as pregnancy-related symptoms or a common cold, might be misdiagnosed was an indication of negative judgments from others. These experiences could reflect a potential broader issue of stigma and discrimination associated with COVID-19. The women’s experiences confirm that the fear of being misjudged as having COVID-19 had significant emotional consequences. Their stories reflected the extent to which the COVID-19 pandemic altered perceptions and practices regarding illness and health in general, with women compelled to navigate the need for professional health care and their fear of being caught in the COVID-19 misdiagnosis net.

#### 3.1.2. Difficulty Using Public Transportation during Pandemics

Against the backdrop of COVID-19, many women struggled to access care. For these women, the journey to accessing care was marred by fear and uncertainty. Their accounts of their experiences encapsulated the emotional and physical stress women went through to access care for themselves and their children. In this subtheme, the challenges women faced with limited transportation because of the impact of movement restrictions provided a window into the profound effect of the COVID-19 pandemic on women. These women’s experiences demonstrated the importance of community support and safety measures in ensuring access to healthcare services for vulnerable groups such as pregnant women and children. For example, Chidinma’s story about her child’s illness and the difficulty they had getting to the hospital, which she was able to solve with the assistance of a stranger, provides insight into the possible support systems in communities during emergencies. Her experience highlighted women’s challenges in accessing healthcare services during the COVID-19 pandemic, especially in areas with limited transportation options. It also emphasizes the importance of community support in overcoming such difficulties.
“*During the COVID-19 pandemic, one of my children came to spend the vacation with me and fell ill. Due to the restrictions on movement, we could not leave the house, so I had to call [on mobile] on health workers for assistance. Initially, I treated him with anti-malaria medication, but when he continued vomiting, I realized we needed to take him to the hospital. Unfortunately, there were no available vehicles in our area, so we decided to walk to the hospital. It was a strenuous journey, but we were fortunate to encounter someone who was allowed to be out during the lockdown and who kindly offered us a ride. We eventually arrived at the health center where the child received good care*”.[Chidinma, Ebonyi]

Chidinma’s first response to self-medicate revealed a self-reliant approach, which gradually gave way to a realization that the condition was severe as the child’s illness progressed. Her sudden shift in perspective transitioned from accepting the limitation of her internal locus of control to acknowledging the need to seek professional medical care. This reflected her ability to adapt and respond to unprecedented challenges. This ability is reflected in her decision to walk to the hospital, which shows her determination to protect and care for her child. Her encounter with the stranger symbolized the human capacity for solidarity in times of crisis. It demonstrated the hope and connectedness of communities in an isolating experience like COVID-19.

Much like Chidinma, Funke’s experiences using public transport during the period were marked by anxiety. This suggested the need for transportation alternatives and safety measures to address women’s challenges in attending ante-/post-natal visits during pandemics.
“*Ehmm, for me, the challenge was that when I went to ante-natal because we did not have a car, I had to take a taxi, and you know how those drivers usually pack us like sardines [tight]. I always pay for two seats, so I do not touch my body with people*”.[Funke, Ogun]

Funke’s statement provided an insight into women’s coping mechanisms and perception of personal space. She described being “packed like sardine” in a taxi, which illustrated not just her physical discomfort but also an emotional unease related to being pregnant and using public transport. The metaphoric use conveyed a sense of overcrowding and lack of personal space, which can be distressing for pregnant women. It was evidently physically discomforting, indicated by her decision to “pay for two seats” to ensure some measure of physical distance from others. Her reaction should be interpreted as an awareness of her heightened vulnerability because of her pregnancy.

Funke’s story showed the additional precautions women in similar situations are often forced to take in public places. This reflected a broader social issue around vulnerable population’s access and comfort in public transport. It highlighted the lack of societal consideration for pregnant women in public transportation and the challenges pregnant women must navigate every day.

#### 3.1.3. Preference for Local Chemists

Similar to Ibironke, some women reported that during the initial stage of the COVID-19 pandemic, restrictions on intra- and inter-state movements compelled them to return to using local chemists. This was a common practice before the COVID-19 pandemic, with some women preferring to consult with a local chemist when seeking healthcare advice. The proximity of the chemist to their community and their success in receiving what they perceived as effective treatment from chemists in the past were significant predisposing factors to using chemists. However, the same women who preferred first to seek care from the local chemist also noted that they would eventually visit a hospital for professional medical services. The below extracts from Esther and Justina highlight the importance of the accessibility and affordability of healthcare services, which can impact women’s health-seeking behaviors.

Esther’s decision to consult a chemist is based on her familiarity with him and the immediate relief she experienced in her prior encounter with him. While she expressed that closeness and familiarity were the significant factors that influenced her decision, her statement also highlighted that the decision was connected to her trust in the chemist. It suggested she was comfortable with the familiar and immediate services she received from the chemist, despite his advice to seek professional care from a hospital.
“*I first took him to the chemist to complain about what was wrong with him, so the chemist told me to take him to the hospital. I took him to the health center. But I still prefer the chemist a little. The chemist is close to us; once we get there and we complain about what is doing us when they write drugs for us, and we use, and we see changes, that is okay*”.[Esther, Sokoto]

In contrast, Justina’s approach to her health appeared more pragmatic and driven by experience. She used the severity and duration of an illness as a benchmark for making her health decision to seek care either at a hospital or a chemist. Her statement showed that she had a nuanced understanding of her health needs and the resources available in the system to support her health journey. Like many women interviewed, Justina showed implicit trust in her decision to seek professional healthcare after a specific illness or ailment surpassed her self-medicating capabilities. Her statement showed some level of empowerment in health-related decision making, likely rooted in her accumulated experience with healthcare.
“*The decision to go to the hospital depends on the severity of the illness. If the sickness lasts for three to four days and becomes complicated, we will go to the health center as quickly as possible. However, if it’s only a two-day sickness, we usually go to the chemist in our area for injections*”.[Justina, Ebonyi]

The women interviewed used certain words or phrases that showed the process or pathway to decision making during the COVID-19 pandemic. For instance, Justina used “severity” to indicate her perception of urgency for healthcare seeking and decision making. The COVID-19 pandemic may have intensified her sense of responsibility and the urgency of seeking care once her 3 to 4-day benchmark was exceeded. Justina’s experience is similar to that of other women in this study. Their ability to navigate the difficulties of accessing timely care during the COVID-19 pandemic hinted at their resiliency and adaptability to the unexpected demands of COVID-19. It showed that the decision to seek care or self-treat was often based on their perception of urgency, which might also be connected to their concerns about safety from COVID-19 exposure.

In the below subtheme, we discuss how the protocols, and their overall perception of the COVID-19 pandemic, influenced their behavior and decision to seek care.

#### 3.1.4. Discomfort with COVID-19 Protocols

This subtheme highlights women’s perception of COVID-19 protocols and the impact on their healthcare seeking during the COVID-19 pandemic. It emphasized the complex interplay of practical challenges, personal autonomy, societal responsibility, and the emotional burden of adapting to new health and safety norms during a public-health crisis. It underscored the importance of public-health messaging and education to dispel fears and encourage women to seek medical attention while following instituted protocols during pandemics to ensure their safety.

Some women expressed challenges in complying with the instructions given by healthcare providers, following the procedures put in place by hospitals, and discomfort with certain aspects of healthcare services. Aisha’s statement illustrates this:
“*Sometimes, going to the hospital can be difficult due to the procedures they have in place. They ask us to keep a certain distance from others, wash our hands, and wear a face mask before entering. It’s not easy to comply with all of these rules*”.[Aisha, Sokoto]

She expressed that receiving instructions from nurses to comply with COVID-19 prevention measures, such as wearing masks and washing hands, was distressing. She expressed difficulties following the protocols, such as maintaining physical distancing, wearing a face mask, and using personal protective equipment. Aisha’s experiences further highlighted the societal and psychological dimensions of her struggle within the broader context of interactions with health workers. She expressed difficulty with the established safety protocols and indicated she was overwhelmed with the requirements for safety. It showed the physical inconvenience and emotional toll the safety protocols had on women like her. Fatima and Iya Pupo further supported her statement in the below extract from their interviews.

Fatima, like Aisha, revealed a complex emotional and psychological response to the healthcare protocols during the COVID-19 pandemic. She felt overwhelmed and intruded upon by the constant health reminders, which she may have perceived as eroding her autonomy. While she acknowledged the need to comply since she had no choice, it reflected her difficulty with balancing her personal freedom with the protocols.
“*The nurses are always bothering us with instructions to wear nose masks, wash hands and do this and that. It can be stressful, but we have no choice but to comply*”.[Fatima, Sokoto]

For Iya Pupo, there was a feeling of uneasiness with how nurses during the COVID-19 pandemic must wear gloves before attending to them to either check blood pressure or administer injections.
“*When they check our ifunpa giga [blood pressure], they wear gloves, but when it comes to injections, we don’t like the way they do it*”.[Iya Pupo, Ogun]

She further stated that the nurses’ conduct is off-putting, “They usually act in a certain way that puts us off”, indicating a potential lack of trust or comfort with the healthcare provider. Such negative experiences may have influenced some women’s willingness to comply with other healthcare instructions, including those related to COVID-19 prevention. Although, as Fatima acknowledged, “It can be stressful, but we have no choice but to comply”. Her statement indicated that she recognized that complying with these measures was necessary.

Aisha and Fatima acknowledged that these measures were implemented for the public’s safety. Yet, their statements highlighted the difficulties of following them, such as maintaining physical distancing, wearing a face mask, and interacting with the health workers using personal protective equipment.

When queried if this experience was unique to COVID-19, Aisha, Fatima, and Iya Pupo confirmed that health providers stopped using gloves while attending to them once restrictions were removed. Aisha’s statement illustrated this—
“*…Since the COVID-19 pandemic stopped, we do not see them covering like before, it is now better*”.

These safety protocols represented a significant deviation from what these women were accustomed to when visiting a hospital, forcing them to adapt to a new normal. From their statements, there appeared to be a tension between compliance and resistance. On the one hand, they acknowledged that the protocols were necessary, which implied they understood the importance of public safety. However, they struggled with the burden of complying with them, which indicated a tension between their personal autonomy and collective responsibility. This sort of tension seemed symbolic of shared experiences such as a global pandemic where a person’s freedom is at odds with the larger public’s safety.

### 3.2. The Influence of Diverse Healthcare Practices and Beliefs on Health-Seeking Behavior

This theme sheds light on the community’s diverse healthcare practices and beliefs impacting healthcare-seeking behavior, with some women relying on traditional or spiritual remedies while others sought professional medical services. Several norms and beliefs fall under this theme, each explained in detail below.

#### 3.2.1. Negative Perceptions of Hospitals and the Apathy of Health Professionals

Several women were reluctant to seek medical attention until their symptoms became too severe to ignore. This hesitance to seek medical attention only when the situation becomes dire creates a barrier to accessing healthcare services. It highlights potential constraints on early detection and intervention. Funke’s statement illustrates this:
“*Like me, I don’t like hospitals. During the lockdown, I only had a headache. Usually, if I have a headache (Ori fifo), I will sleep once I sleep, I am okay, if I now see that my body is not that okay, that it is getting too much, I will come to the hospital to get treatment*”.[Funke, Ogun]

For Funke, who stated, “Like me, I don’t like hospitals. I will sleep once I sleep, I am okay” and only seek medical attention if the symptoms persisted and became unbearable, “if I now see that my body is not that okay, that it is getting too much, I will come to the hospital to get treatment”. This behavior indicates a hesitance to seek medical attention unless the situation becomes dire. This behavior becomes exacerbated during infectious diseases because of the restrictions and potential perceptions of distrust about the health system. Stella’s statement further illustrates this behavior,
“*I don’t usually go to the hospital. What did I want to go and do at the hospital? The Coro (COVID) we know that they just use it to take our money, and when we get to the hospital, they will be doing different kinds of things*”.[Stella, Ebonyi]

The risk of COVID and its implications does not nudge her towards a hospital when sick. Although, in using the phrase “I don’t usually go to the hospital”, she indicates a habitual behavior of not going to the hospital when sick. Stella self-identified as a nurse (she is an auxiliary nurse although unemployed since finishing school), and yet showed a distrust of the health system. Her perspective on the health system may be biased because she has not worked since graduation. Her cynical views of the health system and the response strategy could be due to her struggles with unemployment. As such, her perception of the health system would be clouded by personal prejudices. Iya Popu further illustrates her statement of distrust of the health system:
“*My view regarding health during the COVID (Coro) outbreak is that medical professionals collude with politicians who seek to create problems for us. Therefore, I believe that when things are running smoothly, good health will prevail among us*”.[Iya Popu, Ogun]

Iya Popu’s statement underscores the significance of examining the larger societal and cultural context in which women experienced the COVID-19 pandemic and accessed healthcare. Her statement suggests a perception of the COVID-19 pandemic and the role of medical professionals and politicians shaped by suspicion and mistrust. Iya Popu and Stella share the same distrust for the health system. However, Stella, an unemployed auxiliary nurse, could have a bias against the health system that distorts her view because of her unemployed status. Iya Popu believes that medical professionals and politicians are collaborating to create problems for people. These viewpoints point to a broader societal issue of mistrust in authority figures and the system, which may have impacted healthcare-seeking behaviors during the COVID-19 pandemic. Additionally, Iya Popu’s belief that good health prevails when things are running “smoothly” highlights the significance of stability and predictability in maintaining health. This perspective contrasts with the uncertainty and disruption created by the COVID-19 pandemic, which can increase stress and anxiety that negatively affect health.

#### 3.2.2. Religiosity as a Normative Practice of Health Prevention and Treatment

Other women mentioned that they do not give their children any medication, instead relying on the grace of God and prayer, which provides them with the confidence to live normally, regardless of the presence of COVID-19. This suggests that some women in the community may rely on faith-based or spiritual remedies for their health, which may not always align with scientific or medical practices. Stella’s statement illustrates this behavior:
“*See, as children of God, we will not encounter sickness that we cannot handle*”.[Stella, Ebonyi]

We see her religiosity influencing her behavior and decisions. Iya Beji further illustrates this behavior in the following statement:
“*I don’t give my children anything. It is the grace of God and the prayer water of Ayodele Babalola, the water from Ikeji. COVID or no COVID, we live our life normal*”.[Iya Beji, Ogun]

She confirms her religiosity as a habitual practice for healthcare. In her statement, “It is the grace of God and prayer water”, we see her religiosity gives her the confidence to live normally, much like Stella, regardless of the presence of COVID-19. Her statement could suggest a strong belief in spiritual healing, which is not discouraged within indigenous health. Instead, it is acknowledged as a complementary health practice. Iya Beji’s use of the name of a religious figure could indicate control from an authority figure in her life, resulting in her reliance on the “prayer water”. As such, when using the phrase “COVID or no COVID, we live our life normal”, she is dismissing any potential threat of COVID-19, possibly because of her belief system.

Stella and Iya Beji’s statements have critical significance for emergency preparedness. Women with such beliefs could view established preventive protocol as contravening their religious beliefs, which will increase their exposure to infections and increase the risk of transmissions within a community. However, their perceptions and position raise an ethical dilemma for public health professionals who design plans and processes for managing health emergencies. The dilemma of creating a balance between spiritual healing and biomedical care is also relevant for non-emergency periods. Although, both women differ in their approach to spiritual healing. For Stella, who seems to have a strong opinion of hospitals and health workers, her statement may be a coping strategy to deal with the discomfort of acknowledging the risks of COVID-19. Her statement, “…we are children of God…” could indicate her belief that she is less at risk because of her belief system. Stella’s statement does not suggest that she does not access professional health care. Instead, it indicates that, like Funke, she only uses it when it becomes severe, confirming that the severity of an illness will prompt the intention to seek care. In the subtheme below, we discuss how some women appeared to revert to habitual behavior, i.e., self-medication, during the COVID-19 pandemic.

#### 3.2.3. The Habitual Practice of Self-Medicating

Several women indicated that self-medication was a common practice during the COVID-19 pandemic. They expressed that seeking medical attention from a hospital is not considered essential in their community and is only done when the illness is severe. Aisha’s statement below illustrates this behavior:
“*When growing up in the village, we normally do self-medication. No one is bothered about going to the hospital. We do not see it as something important. We only go to the hospital when it is severe*”.[Aisha, Sokoto]

For Aisha, this approach may seem reasonable. However, it can be taken to the extreme when health-seeking behavior is avoided or delayed due to prejudices against formal healthcare, even when the disease is severe or best served by formal care. Aisha uses the phrase “no one is bothered” to rationalize her decision to engage in self-medication, thus demonstrating cognizance of the existence of appropriate and acceptable healthcare approaches that warrant exploration. However, due to the social norm of this practice, she perceives it to be socially acceptable and continues to engage in such behavior.

In contrast to Aisha and Iya Beji, Chioma, while admitting to reverting to self-medicating during the COVID-19 pandemic, said,
“*Truly, if it is for a minor illness like a headache, “I do not go to the hospital. I do go to the hospital when my child is sick. But, then, every time the child is ill, we do not take them to the hospital because of fear of corona*”.[Chioma, Ebonyi]

She seemed comfortable self-medicating, yet recognized the importance of seeking appropriate medical care when she stated, “I do go to the hospital when my child is sick”. However, due to the fear of contracting COVID, she stated, “But, then every time the child is ill, we do not take them to the hospital because of fear of corona”. Although Aisha, Iya Beji, and Chioma’s position depict self-medication as a favorable societal norm, it also highlights the role of social pressure on amplifying the negative impact of internalized belief systems, which, when consistently repeated over generations, become an acceptable way of life. The following subtheme illustrates how internalized belief systems influenced women’s decision to revert to prior health-seeking behavior.

#### 3.2.4. The Norm of Combining Medical and Traditional Medicine

Several women in the study reported using herbal or a combination of traditional remedies and prescribed medication for their health concerns. This is illustrated by Damilola, who stated that taking a combination treatment seems reliable when she stated, “I used drugs and herbs, both of them are working together for me”. Damilola’s statement suggests she believes that both systems are effective. It also indicates that she is willing to explore different healthcare pathways, which could reflect her past experience using both practices.
“*I used drugs and herbs. Both of them are working together for me. Usually, after three to four days, if we do not see any improvement, we will quickly go to the health center, but for now, we will use the nurse in our area*”.[Damilola, Ogun]

From Damilola’s statement, “after three to four days if we do not see any improvement, we will quickly go to the health center…”, it is evident that reverting to professional healthcare after using the traditional treatment after a few days is an acknowledgement and an understanding of the role of hospitals and health workers. It could also suggest an understanding of the need to initiate timely conventional treatment for ailments if traditional treatment does not work. Due to the COVID-19 pandemic, she visits a nurse in her area, “but now we use the nurse in our area”. Her statement could indicate trust or confidence, given that the health worker is within her neighborhood.

Damilola’s statement does not suggest a preference for traditional remedies. Instead, it reflects the influence of medical pluralism and the internalized belief in the efficacy of alternative medicines. This is further illustrated in Bimbo’s statement:
“*Some pregnant women believe in those [traditional healers] that give them drugs in the house like those trado-medical. They prefer to go there because they believe that herbs and roots are created by God despite the health problems they face after taking some traditional medicines*”.[Bimbo, Sokoto]

Bimbo’s statement reveals a known fact about women and their preference for traditional birth attendants and healers. The belief that herbs and roots “are created by God” indicates the role of religiosity in health-seeking behavior. Her observation of women’s behavior could be from her role as an auxiliary nurse, which would have exposed her to women’s intentions and experiences with traditional medicine. Her statement provides useful insight into the potential repercussions of using traditional medicines and why women use them.

Bimbo’s statements provide insight into Iya Beji and Stella’s religiosity and its influence on their health practice. She confirmed the integration of traditional and professional healthcare with women’s willingness to explore alternative care. However, she showed the differences in health pathways for women. For instance, while religiosity influenced these women, Iya Beji and Damilola, each woman responded differently to it. Iya Beji relied on the power of prayer and holy water, while Damilola used traditional medicine, likely because God created it.

This theme highlighted the varied attitudes of the women interviewed toward healthcare-seeking behavior and the role certain norms and internalized beliefs play in their healthcare-seeking behavior. Many women reported staying home and using home remedies or chemists to treat themselves or loved ones. Others reported resorting to traditional practices as an alternative to hospital visits. In both cases, however, the women who did describe staying home or seeking traditional practices also reported visiting hospitals when their illness or that of a loved one became unmanageable at home. It highlighted the potential for women to adopt unsafe practices when restrictions prevent them from accessing professional healthcare.

### 3.3. Gendered Notions of Responsibility and of Coping with Financial Challenges

This theme explored the women’s coping and financial skills during the COVID-19 pandemic, their decision-making power to secure funds needed for timely care, and their perceptions of financial roles in the family.

#### 3.3.1. Gender Dynamics and Financial Roles

In this subtheme, we explored the relational dynamics of marriage among the women interviewed and the financial role women play or perceive as their role in the family. It also explored their perception of the decision-making path for health. Women’s stories highlighted a social norm requiring women to inform their spouses, not as a choice but as a mandatory marriage obligation. During the COVID-19 pandemic, some women indicated that this norm persisted despite their spouses’ inability to provide financial support. Esther, Fatima, and Ibironke’s statements below illustrate this:
“*I have to tell our husband because it is mandatory. I do so since he is the head of the house. We asked him if he had money to give me, he does, but if not, he would suggest how to get it*”.[Esther, Sokoto]

Esther’s statement, “I have to tell our husband because it is mandatory…”, suggested that she does not have a choice as the culture or society requires that she inform her spouse of issues concerning the house. However, it could also mean that she sees her spouse as a partner and feels he should contribute to the decisions concerning their home. However, her following statement, “I do so since he is the head of the house”, would contradict this, as it alludes to the fact that traditional gender roles in her society have ascribed specific fundamental duties to men and women. As such, she must observe those roles. This is reflected in her statement when she mentions that she will ask her spouse for money and follow his suggestion to acquire it if he does not have it. It reflected that she is financially dependent on him. Her use of the word “mandatory” implied a hierarchical dynamic in their relationship, which is common in patriarchal societies. However, it is unclear from this statement that Esther lacks agency or autonomy in financial matters. Esther’s statement revealed that social norms in place assigned men the financial responsibility for the family. Ibironke’s statement below supports this:
“*Why would I not tell my husband? He is my oga na! Ah, if my child is sick, I will tell him the child is sick; even if he is travelling, I will call and tell him. He will ask me to borrow the money, and he will pay when he returns. I will go to my brothers; they will always give me*”.[Ibironke, Ogun]

For Ibironke, her spouse is unquestionably her support system. Her statement revealed confidence in her relationship with her spouse as her provider. Her rhetorical question, “Why would I not tell my husband? He is my oga na!” She used the word “Oga”, often used in Nigeria, to depict a person of authority to describe her spouse, suggesting a hierarchy in her relationship like Esther. While her statement revealed her financial dependence on her spouse, it also showed her perception of the importance of spousal communication. However, it could also be interpreted as her inability to make a decision without spousal input when he is away, “I will tell him the child is sick, even if he is travelling, I will call him and tell him”. Her statement further revealed an evident dependence on her siblings that appears to be an integral part of her life. Ibironke identifies as a businessperson like Esther, and there is no indication that she does not have the autonomy or agency to earn her own money. Like Ibironke and Esther, Fatima’s statement revealed her inability to function without her spouse.
“*During that time [COVID-19 restriction intrastate movement], my child was sick, and the father travelled then, so taking care of myself was difficult. I borrowed money from my neighbor and then paid it back when the father returned*”.[Fatima, Sokoto]

Her statement indicates that she is financially dependent on her spouse. However, she can decide to seek assistance from a neighbor and pay this back when her spouse returns home. She demonstrated poor resiliency to cope with situations or personal emergencies without her spouse. Her statement, “…taking care of myself was difficult…”, revealed that her difficulty was not about her child’s illness or her well-being but also that she is unable to cope with difficult situations without her spouse. Fatima’s inability to care for herself without her spouse suggests a much broader dependency that goes beyond financial needs and potential vulnerability during emergencies.

Ibironke, Esther, and Fatima’s statements demonstrated their dependence on their spouses for financial support. They may also indicate a cultural or social expectation of husbands as primary financial providers. The women’s statements suggest a practice where gender roles are traditional, with each gender assigned to roles in society. For instance, Ibironke and Esther used phrases like “oga” and “head of the house” to show the elevated status of their spouses, possibly with more decision-making power. The choice to involve their spouses even when they are not in the same location could be interpreted as the women’s choice. It also reflects an understanding of the importance of communication among couples about finances.

In contrast, each woman has varying degrees of autonomy and perceptions of their spouse’s role in the family. For instance, on the one hand, Esther implied she had little to no choice in consulting her spouse, unlike Ibironke and Fatima. In the next subtheme, we present women’s alternative approaches to seeking financial support during the COVID-19 pandemic.

#### 3.3.2. The Reliance on Extended Family as a Social Network

Unlike Fatima’s statement, “I borrowed money from my neighbor and then paid it back when the father returned”, some women stated they would only reach out to close family members. These women felt that often asking for help from people outside of their immediate family would reduce their social image. More specifically, women felt that asking outsiders for help, including friends, could result in petty gossip, which in small communities tends to be stressful. Chioma and Halima’s statements illustrate this:
“*That time, people did not have. I am the sole breadwinner for my family. My husband did not have a job because, during the COVID, nobody was digging boreholes. I always ask my siblings, but will manage if they do not have them*”.[Chioma, Ebonyi]

Chioma’s statement indicated some economic pressure during the COVID-19 pandemic, given her husband, who drills boreholes for a living, was out of work. Her statement suggests a reversal of her traditional role when she says, “I am the sole breadwinner”, which could add a layer of economic stress. Her statement on her husband’s employment status revealed the effect of the COVID-19 pandemic on the informal sector. It provides a window into the struggles of similar families, dependent on the informal sector for income. Chioma revealed seeking assistance from her siblings or others within her family in times of crisis. However, she uses the phrase “if they do not have, I will manage” to indicate that she will manage should they fail to assist. Chioma’s story reflects the resiliency of women who had to carry the additional burden of supporting their families during the COVID-19 pandemic while providing emotional support to their children:
“*At that time, we did not have food. When I asked my brothers for help, they also did not, so I went home. I used the small change I had to buy garri [fried cassava flakes] and groundnut [Peanut]. It was very small, and I soaked it so it would swell. We didn’t even have sugar, but I used salt. When it was soaking, I called my children, and we began to sing around the garri until it swelled. The singing made it fun for my children. We ate and were full*”.[Chioma, Ebonyi]

Chioma’s resiliency and ability to adapt to the crisis and ensure her children do not feel emotional stress suggests a strong emotional resiliency and resourcefulness, which enables her to turn a dire situation into a memorable experience for her children, shielding them from their situation.

For Bimbo, her spouse was the first person mentioned when asked whom she went to for financial support during the COVID-19 pandemic, like Ibironke, Esther, and Fatima. Going to her extended family will only happen if her spouse does not have it.
“*If my husband does not have to give, I will ask my mother and my sisters; if they do not have, I will not ask my friends or neighbors. You know how people talk. Tomorrow, they will say, so, and so is suffering and is even borrowing money from everyone*”.[Bimbo, Sokoto]

However, unlike Fatima, who is comfortable asking her neighbors, Bimbo emphasized her concern about how her neighbors will perceive her asking for financial assistance during the period. It suggested a fear of revealing her family’s circumstances to others outside her immediate family. It could also be that there is some form of stigma attached to asking for money from neighbors, which could result in people talking about her family as “suffering”. Bimbo identifies as unemployed, a critical aspect of her story because it places her as being financially dependent, making her vulnerable like Fatima. Her hesitation in asking for assistance from her neighbor could be a mental coping mechanism as well as her need to maintain some illusion of normalcy in her family.

While Chioma and Bimbo dealt with financial difficulties during the COVID-19 pandemic, Chioma was a primary provider for her family, and Bimbo was an unemployed woman. They indicated that the support of their families in times of crisis is crucial. From them, it is obvious the role familial networks play in ensuring the resiliency of women during environmental disruptions like the COVID-19 pandemic. Both women approach their financial challenges similarly, although their reasons for taking the approach may differ. For instance, Chioma is unwilling to burden others with her problem, choosing to care for her family with the little she has, and Bimbo is reluctant to expose herself to gossip.

## 4. Discussion

This study aimed to understand women’s behavior while seeking care during the COVID-19 pandemic and where and why they chose to seek or not seek care. These findings contribute to a growing body of literature on women’s experiences during the COVID-19 pandemic’s early days and, in general, women’s experiences accessing health care during infectious disease events [9,38,39,40,41,42,43,44,45,46,47,48,49,50,51,52,53,54]. To the best of our knowledge, this is the first interpretative phenomenological analysis of these experiences in Nigeria.

Our study expands and also challenges existing models of health-seeking behavior by highlighting how the COVID-19 pandemic altered known behaviors, deviating from the influence of cultural and social norms, previously considered predominant factors in health-seeking behavior. These models often highlight socioeconomic, cultural, and environmental factors as the key drivers of women’s health-seeking behavior. However, findings from this study show a more nuanced perspective from the women’s view, which focuses on the unique impact fear had in women’s behavior. It is crucial, as it suggests that health strategies need to address fear and anxiety as core elements in promoting health-seeking behavior during pandemics. For instance, in demonstrating that culture and social norms did not play a significant role in women’s health seeking during the COVID-19 pandemic, this study suggests that in the face of uncertainties and health risks, women will prioritize timely and safe healthcare over the expectations of culture and social norms.

According to the WHO [49], the disproportionate impact of the COVID-19 pandemic on women increased gender inequities as a result of peculiarities, stringent social norms, and gender roles, threatening women’s empowerment and well-being in many countries, including Nigeria. Other studies on COVID-19 showed that these peculiarities, which often present as culture, significantly affected women’s agency, health decisions, and behaviors [55,56,57]. From the findings of this study, social norms or culture did not play a role in women’s decision making during the pandemic. Instead, the challenges and managing the risks of COVID-19 were significant drivers of women’s health-related decision making. Similar studies on the impact of COVID-19 on women suggest there is no direct connection between social norms and culture and women’s decision making during the pandemic; instead, the overall challenges and risks associated with the COVID-19 pandemic influenced their decisions [58,59].

In this study, our results demonstrate the complexity of navigating healthcare during the COVID-19 pandemic. Women experienced barriers to seeking or accessing healthcare for themselves and their children largely due to the fear of contracting COVID-19 at health facilities or in transit. However, they also encountered other barriers, such as financial constraints and limited access to transportation. This finding on fear contradicts a similar study on health-seeking behavior during the COVID-19 pandemic, which found that fear did not affect women’s health-seeking behavior [60]. However, in this study, fear influenced women’s health practices during the period under study, with women engaging in different health practices like self-medicating.

These self-medicating behaviors, our women asserted, existed before the COVID-19 pandemic. Similar studies in Bangladesh and Ecuador found that self-medication was a response to the lockdown, isolation, concerns about the virus, and the infodemic during the COVID-19 pandemic [61,62]. In contrast, another study found that the percentage of women who reported their self-medicating during the COVID-19 pandemic was similar to that of the pre-COVID-19 pandemic [63]. Nonetheless, our results indicate that pandemic-related restrictions on movement and lockdowns likely exacerbated these women’s self-medicating behavior patterns. Since self-medication for minor ailments is rational and encouraged in all settings as it represents the most efficient and cost-effective management of most minor illnesses, it is the extreme and inflexible interpretation of self-reliance which may lead to harm and not the issue of self-medication that is itself to be questioned overall. For women who chose to visit the hospital, there were difficulties securing transportation, with some often resorting to travelling long distances on foot, as shown in Chidinma’s story. However, from the women’s statements, COVID-19 protocols created additional barriers at the hospital. A study in Brazil also made a similar finding that changes in maternal practices at the hospital affected women’s mental health and well-being [64]. This finding highlights the need for a more empathetic and accommodating public infrastructure that considers the impact preventive measures could have on creating negative perceptions of the health systems and health workers.

Another key influence on women’s health practices during the COVID-19 pandemic was the reliance on religion and spiritism as alternative preventive and treatment measures. For instance, some women believed the COVID-19 pandemic was an illness only treatable by prayer. In Nigeria, as in numerous other African nations, religion and spirituality are deeply embedded in people’s daily lives, constituting a significant factor in the perception and management of health-related issues. This cultural orientation has played a vital role in guiding people’s responses and coping mechanisms to the challenges of the COVID-19 pandemic. This belief may be linked to early religious predictions on the outcome of the COVID-19 pandemic touted by religious leaders in Nigeria who refused to adhere to the restrictions and subsequently [65,66,67,68]. This finding contradicts a study in southwest Nigeria that reported that while spiritism influenced health-seeking behavior, participants did not perceive the COVID-19 pandemic as an act of God [68]. This attitude towards the COVID-19 pandemic was not unique to women in Nigeria. Globally, there was evidence of a religious stance against COVID-19 as an illness not meant for “true” believers of God [69].

In Nigeria, the notion that the government manipulates the public for political gain is deeply entrenched within the country’s social fabric. As governments attempted to persuade people about the severity of the COVID-19 pandemic, women may have perceived a conflict between the messaging and the general attitude of those they regarded as authority figures. The history of distrust between citizens and politicians in Nigeria [70,71] could directly reflect the general distrust of the health system among citizens. This distrust of politicians, who frequently serve as the face of pandemic responses, may result in a lack of confidence in the health information dispensed by the government and a reluctance to seek necessary healthcare services. Women’s health, including access to reproductive and maternal healthcare services and increased susceptibility to infectious diseases, could be adversely impacted as a result. Studies have found that mistrust of the government affected the adoption of COVID-19 protocols, increasing its prevalence rate [72,73,74,75].

Additionally, given the history of widespread mainstream media manipulation in Nigeria, people continue to turn to social media for information mainly due to their distrust of mainstream media, which they would perceive as being government sympathizers [76,77]. The volume of misinformation penetrating communities from such social platforms has been shown to harm people’s reactions to the COVID-19 pandemic. According to several studies, one of the identified issues is the velocity of misinformation and disinformation circulated at the onset of the COVID-19 pandemic [38,78,79,80]. Much of this misleading information was reiterated over digital media, with many religious leaders misleading congregations on its actual intensity. The framing of the COVID-19 messaging may have also affected how women received it. For instance, studies found that early messaging on the origin and source of the COVID-19 pandemic affected the perception of severity globally [38,79].

Future research could explore the impact of globalization on gendered and social roles in Nigeria. It could also examine the role of messaging on women’s health literacy and health-seeking behavior. This topic is essential as it highlights the importance of transparency in promoting health behaviors and addressing health disparities. It is even more important now, given the rise in misinformation about diseases and ailments, because of the expanding influence of technology.

Additionally, women’s ability to leverage technology during this period indicates the potential of telemedicine in increasing access to healthcare. For example, Chidinma called a nurse for assistance when her child was sick. There would be merit in a study exploring digital technology’s role in increasing women’s healthcare access during pandemics. Finally, it is essential to examine health workers’ mental health during the COVID-19 pandemic, given their increasing burnout and stress and women’s perception of their role in the negative health experiences reported in this study. With the global shortage of health workers estimated to continue over the long term, it is essential to understand the needs of health workers during future pandemics and other shocks to the health system.

While this study contributes to existing reports on women’s health experiences and perceptions of their health-seeking behavior during infectious disease outbreaks and pandemics, it has some limitations. The women’s recruitment was carried out through local research assistants resident in the study location. All the research assistants employed in the study already have established relationships with the communities where the fieldwork for this study happened; as such, these may have influenced the dynamics and findings of the result as women may give responses that they felt would be acceptable to them. Additionally, the lead researcher as an outsider may have affected the study results as women may have withheld vital information that could provide robust data for analysis. However, the lead researcher switched roles with the research assistants once this was noticed to avoid any potential data contamination.

To mitigate the risk further, the lead researcher kept a field journal, which included daily reflective diary entries concerning field activities. The journal supported the lead researcher in reflecting critically and actively on the information shared by the women. It was also helpful as a reflective tool for the lead researcher to examine her pre-existing biases, pre-conceived beliefs, and assumptions and acknowledge her positionality better to understand the phenomenon of women’s health-seeking behavior and provide a thoughtful and nuanced interpretation. While the lead researcher piloted the tool, the study tools were only piloted in one location because of time and associated costs. Given the heterogeneous nature of Nigerian society, it would have been advantageous to pilot the study tool in the three geopolitical zones of the study location. Piloting them would have ensured the clarity and appropriateness of the tool for women in the various locations. However, given the time and financial constraints, this was not possible. We did not aim to generalize findings across all Nigerian women but to facilitate an understanding of potential social and cultural influences on healthcare-seeking behavior among women navigating infectious disease outbreaks in culturally distinct states. A broader qualitative study across the various ethnic groups in Nigeria is recommended in the future.

## 5. Conclusions and Recommendations

This paper explored where women accessed health care during the COVID-19 pandemic and the mediators of their health-seeking behavior. While this paper focused on where women accessed healthcare during the COVID-19 pandemic, it also explored the underlying reasons for their health-seeking behavior and the specific types of care they received. By doing so, this paper provides a comprehensive understanding of the factors influencing women’s access to healthcare during the COVID-19 pandemic. While the COVID-19 pandemic may have influenced women’s health-seeking decisions, the findings reinforce that certain habitual behaviors persist despite women knowing the implications of such behaviors. There were no significant differences in women’s experiences across the three states of Nigeria. Even where women appeared to approach health-seeking differently, their responses to the challenges resulting from the COVID-19 pandemic were similar.

Several policy recommendations emerging from the data should be considered as countries prepare for the next global health emergency. Below, we outline some critical policy recommendations and implications for consideration that are relevant to Nigeria and the broader global health context.

The proliferation of modern technology in healthcare has been extensively discussed in the literature since the COVID-19 pandemic [81,82]. These studies exploring the integration of the metaverse, augmented reality (AR), and other digital technologies into healthcare highlighted their transformative potential for healthcare demand and supply. With this potential, healthcare can explore innovative approaches to providing services, such as enhancing community-based healthcare services to make them more accessible during pandemics. These innovations could integrate existing telehealth approaches from remote prescribing, drug-dispensing kiosks, remote consultation [83,84], and disease surveillance in one place. To do this will require (1) a multisectoral collaboration between the public and private sectors, including telecommunication companies, and (2) a national strategy focused on expanding the mHealth initiatives to rural areas; leveraging telecommunication companies’ reach will ensure access to the remotest of places.

In this study, religion and spirituality were women’s coping mechanisms during the COVID-19 pandemic. It is important to recognize the enduring influence of religion and advocating with religious leaders to provide messaging that acknowledges and supports the separation of spirituality and access to health care. In policies and programming, it is crucial that physical healing is recognized as a biological or physiological process separate from spiritual healing. In programming, this distinction should be incorporated into health workers’ in-service and pre-service training manuals. It will prepare health workers to provide services that respect women’s spiritual beliefs while emphasizing the complementary role of bio-medical care. Physical healing is a tangible, measurable process through evidence-based medical interventions.

In contrast, spiritual healing is subjective and involves a person’s practices, beliefs, or connection to a higher power, such as mediation, prayer, or other religious or spiritual rituals. Spirituality must not be confused with mental health, which is rooted in biological existence. Spiritual healing is about respecting the diversity of cultures and people’s personal beliefs, allowing for a more integrated approach to health and well-being. As such, the recommendation for a practical approach to health policy and programming is to acknowledge the differences between physical and spiritual healing and to treat them as distinct yet complementary aspects of health and well-being. Before the next pandemic, policies and plans must incorporate spiritual healing as a complementary approach to managing risks. This approach will involve creating community linkages with religious networks to amplify women’s ability to cope with the stressors associated with any environmental disruption. Most importantly, it addresses an ongoing concern in women’s health-seeking behavior and the over-reliance on prayer houses for care and delivery, which in many instances may lead to death.

The public-health approach to messaging often takes a top-down approach, focused on providing factual content and evidence to support medical accuracy and guidance. This conventional approach uses mass media channels to broadcast messages. However, in developing these messages, community perspectives are often under-considered. A more unconventional approach to consider ahead of the next pandemic is engaging women in designing and developing health campaigns. Essentially, it involves taking a more participatory, community-oriented strategy where women lead the development of the messages. This idea is based on co-creation, which is not new in healthcare. Studies have shown that healthcare practices improve when patients are involved in designing their own care approach [85,86,87]. As such, including the women in developing these health messages will begin to address their perceptions, i.e., the fear that anyone presenting to a hospital as ill will be misdiagnosed during an outbreak because of misinformation, which has been proven to limit health literacy [88]. Engaging women from diverse backgrounds to understand their concerns and experiences during the COVID-19 pandemic in a space where they can openly share their challenges would improve future health campaigns. From these sessions, a messaging approach can be developed and refined with feedback from the women. However, while a feedback system must be set up to assess the effectiveness of the health messages, it is important to not reduce the involvement of women in this process to only providing feedback. A continuous and persistent engagement with women must continue throughout the implementation of the messaging campaign.

## Figures and Tables

**Table 1 ijerph-21-00265-t001:** Participant demographics.

**Item**		**#**
Education		
	Secondary school	9
	University	2
	College, i.e., HND, diploma	3
Population area		
	Semi-Urban	8
	Rural	6

**Table 2 ijerph-21-00265-t002:** Participant list.

**Name**	**Age**	**Job**	**Location**
Chidinma	19	Businesswoman	Ebonyi
Chioma	31	Businesswoman	Ebonyi
Stella	42	Nurse (auxiliary)	Ebonyi
Justina	27	Businesswoman	Ebonyi
Damilola	26	Caterer	Ogun
Ibironke	26	Businesswoman	Ogun
Iya Beji	30	Businesswoman	Ogun
Iya Pupo	47	Businesswoman	Ogun
Funke	24	Student	Ogun
Bimbo	23	Nurse (auxiliary)	Sokoto
Aisha	26	Unemployed	Sokoto
Esther	35	Businesswoman	Sokoto
Halima	20	Unemployed	Sokoto
Fatima	22	Unemployed	Sokoto

## Data Availability

The tools and de-identified data presented in this study are available on request from the corresponding author. The data are not publicly available due to privacy concerns.

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
