# Peer review of "“We Live Our Life Normal”: A Qualitative Analysis of Nigerian Women’s Health-Seeking Behavior during the COVID-19 Pandemic"

_ijerph, 2024, doi:10.3390/ijerph21030265_

Round 1

Reviewer 1 Report (Previous Reviewer 1)

Comments and Suggestions for Authors

General comments: the author should note that the manuscript already contains lots of texts and the revision should not further increase the length of the manuscript.

Introduction

  • Missing Context: The introduction effectively sets up the study's rationale but lacks a comprehensive review of existing literature on health-seeking behaviors during pandemics in African contexts, especially comparative insights across different countries. Expanding this section could position the study within a broader discourse.
  • Specific Objectives: While the study objective is mentioned, it would benefit from being more specific about the theoretical frameworks guiding the analysis. For example, mentioning theories related to health behavior change could contextualize the objectives further.

Methods

  • Participant Selection: The manuscript describes the participant selection criteria and recruitment strategies but lacks detail on the diversity of participants in terms of socioeconomic status, education levels, and urban vs. rural residency. Clarifying these aspects would strengthen the study's representativeness.
  • Data Collection Methods: While interviews and focus groups are mentioned, the manuscript could benefit from a more detailed description of how these methods were adapted to COVID-19 restrictions (e.g., use of digital platforms for interviews).
  • Ethical Considerations: The manuscript mentions ethical considerations, but specific details about the informed consent process are lacking. Adding information on how consent was obtained (verbally or written) and how participants' anonymity was preserved during data analysis would enhance transparency.

Results

  • Theme Development: The manuscript presents insightful themes but could benefit from a deeper exploration of how these themes intersect with broader social determinants of health. For instance, discussing the role of gender and power dynamics in health-seeking behavior would add depth.
  • Direct Quotes: While quotes are used to support themes, some themes could be enriched with additional quotes to provide a more nuanced understanding of participants' experiences.

Discussion

  • Link to Literature: The discussion provides a good overview of the findings in relation to the study objective. However, it could be strengthened by more explicitly linking the results to existing theories and studies on health-seeking behavior, especially those focusing on gender and cultural contexts.
  • Policy Implications: The manuscript offers recommendations for policy and practice, which is commendable. Expanding this section to include specific, actionable strategies for healthcare providers and policymakers to address the barriers identified in the study would be valuable.
  • Limitations: The manuscript acknowledges limitations but could further discuss the implications of these limitations on the study's findings. For example, exploring how the qualitative nature of the study limits the generalizability of the results and how future research could address these gaps.

Conclusion

  • Broader Implications: The conclusion succinctly summarizes the key findings but could further articulate the broader implications for global health, particularly in the context of preparing for future pandemics. Highlighting the importance of incorporating gender-sensitive approaches in pandemic preparedness could offer a significant contribution to the field.

Overall Assessment

The manuscript provides important insights into the health-seeking behaviors of Nigerian women during the COVID-19 pandemic. Strengthening the connection to broader theoretical frameworks, providing more detail in the methods section, and deepening the analysis in the discussion could enhance the manuscript's contribution to the literature. The recommendations provided aim to support these enhancements.

Comments on the Quality of English Language

Minor English editing needed

Author Response

Dear reviewer 1:

Thank you very much for the detailed and constructive feedback on our paper. Your comments and suggestions strengthen the overall quality of the paper. We appreciate your taking the time to review and provide these feedbacks. Please find our response to your comments below;

Introduction

Response

  • Missing Context: The introduction effectively sets up the study’s rationale but lacks a comprehensive review of existing literature on health-seeking behaviors during pandemics in African contexts, especially comparative insights across different countries. Expanding this section could position the study within a broader discourse.

Thank you. We have expanded our citations to COVID-19 references in Africa and globally.

  • Specific Objectives: While the study objective is mentioned, it would benefit from being more specific about the theoretical frameworks guiding the analysis. For example, mentioning theories related to health behavior change could contextualize the objectives further.

Our analysis is guided by a feminist intersectional and social theory lens recognizing the plurality of women’s experiences. Using one specific health-seeking behaviour model homogenizes these women’s experiences, which does not fit into our broad study objectives. However, we recognize that each woman’s behaviour pattern can be explained with various HSB models. In our introduction and discussion, we contextualize it within various HSB models.

Methods

  • Participant Selection: The manuscript describes the participant selection criteria and recruitment strategies but lacks detail on the diversity of participants in terms of socioeconomic status, education levels, and urban vs. rural residency. Clarifying these aspects would strengthen the study’s representativeness.

Thank you. Tables 1 & 2 detail the socioeconomic status of our participants. However, we did not intentionally focus on the women’s educational level or employment status because it would exclude certain women from the study, but we were attentive to it. As such it was not a criteria in this study

  • Data Collection Methods: While interviews and focus groups are mentioned, the manuscript could benefit from a more detailed description of how these methods were adapted to COVID-19 restrictions (e.g., use of digital platforms for interviews).

Data was collected after the lockdowns and restrictions were lifted between August and November 2022. As such, it was face-to-face. We did not use any digital platforms for data collection.

  • Ethical Considerations: The manuscript mentions ethical considerations, but specific details about the informed consent process are lacking. Adding information on how consent was obtained (verbally or written) and how participants’ anonymity was preserved during data analysis would enhance transparency.

Thank you. All names in this study are aliases. The women in this study chose the alias themselves after requesting that their names not be used during the focus group and one-on-one interviews. During the focus group, one woman commented that they have participated in many studies and constantly worry that their names and pictures are used. We asked that they use aliases to assure them we would not use their real names.

Results

  • Theme Development: The manuscript presents insightful themes but could benefit from a deeper exploration of how these themes intersect with broader social determinants of health. For instance, discussing the role of gender and power dynamics in health-seeking behavior would add depth.

We acknowledge the importance of contextualizing our findings within a broader social determinants of health framework to deepen the understanding of participants’ experiences. However, we wish to address the caution required when interpreting themes of research participants’ subjective experiences. While we recognize the significance of examining gender and power dynamics in health-seeking behavior, it is vital to remain committed to the participants’ narratives and avoid imposing interpretations that may not accurately reflect their experiences. The theme, Gendered notions of responsibility and coping with financial challenges, emerged from the narratives provided by our participants, predominantly focusing on their coping mechanisms and responsibilities within the context of financial challenges. While gendered aspects may influence these dynamics, we must highlight that our interpretation was grounded solely in the participants’ words and experiences. We would also like to comment that making assumptions about power dynamics without explicit statements from the participants risks misrepresenting their agency and resilience in navigating challenging circumstances. We must be cautious not to perpetuate stereotypes or assumptions about women’s lack of power, especially when the participants themselves do not articulate such sentiments.

  • Direct Quotes: While quotes are used to support themes, some themes could be enriched with additional quotes to provide a more nuanced understanding of participants’ experiences.

Thank you. We acknowledge the importance of enriching themes with firsthand accounts to provide a nuanced understanding of their experiences. However, it is crucial to note the methodological framework of interpretive phenomenological analysis (IPA), which prioritizes in-depth exploration of individual lived experiences within a small sample size. This paper reports on 14 participants, and each participant’s narrative was meticulously analyzed to identify meaningful themes reflective of their unique perspectives. While we recognize the value of additional quotes in enhancing the richness of our findings, the depth of analysis inherent in IPA necessitates prioritizing quality over quantity when selecting excerpts that best encapsulate the essence of participants’ experiences. Therefore, while we have strived to balance the inclusion of direct quotes to support our themes, maintaining the integrity of participants’ voices has influenced our decision regarding quote selection.

Discussion

  • Link to Literature: The discussion provides a good overview of the findings in relation to the study objective. However, it could be strengthened by more explicitly linking the results to existing theories and studies on health-seeking behavior, especially those focusing on gender and cultural contexts.

Thank you. We linked to recent studies on COVID-19 and its impact on health-seeking behavior, including gender-specific studies during COVID-19 in diverse populations in Africa and similar contexts globally.

  • Policy Implications: The manuscript offers recommendations for policy and practice, which is commendable. Expanding this section to include specific, actionable strategies for healthcare providers and policymakers to address the barriers identified in the study would be valuable.

Thank you. Our policy recommendation focused on suggestions we felt were critical considerations at a strategic level and had implications for the next pandemic. As such, our recommendation focuses on policymakers and their ability to revamp or incorporate new dimensions into existing policies. Our actionable strategies are included in the recommendations.

  • Limitations: The manuscript acknowledges limitations but could further discuss the implications of these limitations on the study’s findings. For example, exploring how the qualitative nature of the study limits the generalizability of the results and how future research could address these gaps.

Thank you. We did acknowledge this limitation in the methods section. We have now moved it to constraints with a recommendation for a broader study in the future.

Conclusion

  • Broader Implications: The conclusion succinctly summarizes the key findings but could further articulate the broader implications for global health, particularly in the context of preparing for future pandemics. Highlighting the importance of incorporating gender-sensitive approaches in pandemic preparedness could offer a significant contribution to the field.

Thank you. We agree. Gender-sensitive approaches are at the core of our recommendation. For instance, incorporating digital tools into health systems that give women access to essential health services during emergencies and co-creating messaging with them point to the importance of gender-sensitive approaches that put the needs of women at the forefront of pandemic preparedness.

Overall Assessment

The manuscript provides important insights into the health-seeking behaviors of Nigerian women during the COVID-19 pandemic. Strengthening the connection to broader theoretical frameworks, providing more detail in the methods section, and deepening the analysis in the discussion could enhance the manuscript’s contribution to the literature. The recommendations provided aim to support these enhancements.

Thank you for your detailed recommendations, which have greatly improved this paper. We are committed to balancing incorporating your recommendations and ensuring the manuscript remains focused on its objective. We have linked our results to more existing literature on COVID-19.

Reviewer 2 Report (Previous Reviewer 2)

Comments and Suggestions for Authors

The revised manuscript addresses the remarks I made on the previous submission. Also, the new article is much more comprehensive and cogent than the previous one.

Author Response

Dear Reviewer 2,

Thank you very much for confirming this version meets your expectations. Thank you again for providing a detailed review to strengthen the paper.

Best

Mary

This manuscript is a resubmission of an earlier submission. The following is a list of the peer review reports and author responses from that submission.

Round 1

Reviewer 1 Report

Comments and Suggestions for Authors

General comments The manuscript provides valuable insights into Nigerian women's health-seeking behaviors during the COVID-19 pandemic. However, there are several areas where improvements are needed: Literature Review: The review could be more comprehensive, including recent studies directly related to the pandemic's impact on health-seeking behaviors in similar contexts. Methodological Justification: More detailed justification for the choice of IPA is needed. Why is this method particularly suited to this study, and how does it enhance the understanding of the topic? Participant Selection and Diversity: The process of ensuring a diverse and representative sample is not clearly articulated. How were participants selected, and what measures were taken to ensure a range of experiences were captured? Translation and Validation of Tools: The process of translating interview guides and ensuring their validity in different local languages is crucial, especially in a multi-lingual context like Nigeria. This process should be described in more detail. Depth of Analysis: The analysis could be deepened to draw more nuanced conclusions. How do the findings relate to existing theories or models of health-seeking behavior? Language and Clarity: Conduct thorough language editing to enhance the manuscript's clarity and readability. Other specific  comments

Introduction:

Example: The introduction references general statistics about maternal deaths in Nigeria (lines 33-34) and the impact of the pandemic on healthcare systems (lines 37-39), but it lacks specific references to studies on health-seeking behaviors during the COVID-19 pandemic in Nigeria. This omission misses an opportunity to contextualize the study within the existing body of research.

Cited References:

Example: References such as Ford et al. (2019) and Solanke et al. (2019) (lines 564-569) focus broadly on maternal decision-making and contraceptive needs, which, while relevant, do not directly address health-seeking behaviors during pandemics. More recent and focused studies would strengthen the relevance of the literature review.

Research Design:

Example: The manuscript mentions the use of IPA (lines 90-91) but does not elaborate on why this method is particularly suited for exploring women's experiences during the pandemic, as opposed to other qualitative methods that could have been employed.

Methods:

Example: The participant selection process is described in a general manner (lines 121-138), but specific criteria for diversity (e.g., socio-economic status, educational background) are not detailed. Additionally, the translation process of the interview guides is mentioned (line 118), but there is no discussion on how the translations were validated for accuracy and cultural appropriateness.

Results:

Example: The results section (lines 182-396) presents themes such as "Barriers to Seeking Timely and Appropriate Health Care" but does not deeply analyze how these findings connect to or diverge from existing models or theories of health-seeking behavior.

Conclusions:

Example: The conclusions (lines 507-531) reiterate the findings but do not offer specific policy or practice implications based on these insights, missing an opportunity to contribute to the discourse on healthcare access during pandemics in Nigeria.

Quality of English Language:

Example: Phrases like “women were recruited as participants until 24 interviews were completed” (line 138) could be more clearly phrased as “The recruitment of participants continued until 24 interviews were conducted.

Comments on the Quality of English Language

Moderate English editing needed

Author Response

Dear Reviewers:

Thank you for the detailed review of our article. Your comments and feedback improved the analysis and presentations. Find specific responses to your comments below.

Reviewer 2 Report

Comments and Suggestions for Authors

This is a well-written paper on an important subject, and about a population that would be difficult to reach in another way than the qualitative method used. Relevance, method and results are (mostly) clearly presented.

I would have recommended acceptance after minor revision except for one issue. The paper contains traces of having been extracted from a more comprehensive text (or so it seems). This is most clear in lines 413-417 in the Discussion, which say that "our results demonstrate that ... women experienced barriers ... financial constraints and lack of transport. " But in the results these barriers are not (or hardly) mentioned. Similarly line 421 "restrictions on mobility", "significant distances on foot" and line 290: "The above extracts centred on the financial practices of women". I did not see those extracts. l. 210 "four overarching themes": but only two are mentioned in this paragraph and discussed in the remainder.

Either those barriers should be adequately discussed in the Results, or the sentences quoted should be removed or rewritten. If barriers due to lack of financial means or transport are part of another text, and therefore are not the subject of the present paper, there should be references to that text, and the importance of the social and cultural barriers relative, which are clearly the subject of the present paper, to those more structural barriers should be briefly discussed.

Minor issues:

- l. 35 "the pandemic" please specify. In 20 years we may not be sure which pandemic this article is about.

- lines 75-88: was only one community in each state selected, or more? If more, how many? How were these "targeted". "culturally distinct states": in what way? Please specify briefly.

- lines 104-106. There seems to be a contradiction between "themselves or their children" and "asked participants about their behaviour ... when their child was sick".

- l. 138 please give the distribution of the participants across the 3 states, here or in Table 1, or as a column in Table 2. It is striking, by the way, that location is never mentioned in the results, even though it is an essential part of the selection procedure. Was there no clear difference between locations? Please discuss this point  briefly.

- l 185 Was it to be expected that all women had at least some secondary education, or does this mean that the sample was relatively well educated, relative to the population of women 18-40 in Nigeria?

- 3.1.2 Discomfort with COVID 19 protocols. One wonders what Bimbo and Stella, who state nursing as their profession, thought about the complaints about nurses.

- l. 264 "PPE" what does this acronym stand for?

- ll 523-524 "physical healing is acknowledged as a biological or physiological process separate from spiritual healing". Very true, but this sentence surprised me as it seems to go against the tendency in much health literature to underline the mutual influence of physical and mental health on each other (even though mental is not the same as spiritual). So some further explanation might be useful.

Comments on the Quality of English Language

The text is fine, very readable, but there are a few typos:

- l 46: "pandemic's one-size-fits-all approach". Do you mean the reaction to the pandemic?

- l 372 "contradicting COVID". contracting?

- l 472 "examine the role of massaging on women's health" Perhaps that also, but I guess you mean messaging here.

Author Response

(The authors gave the same response as above.)
